# Oxygen Nanocarriers for Improving Cardioplegic Solution Performance: Physico-Chemical Characterization

**DOI:** 10.3390/ijms241210073

**Published:** 2023-06-13

**Authors:** Maria Tannous, Gjylije Hoti, Francesco Trotta, Roberta Cavalli, Takanobu Higashiyama, Pasquale Pagliaro, Claudia Penna

**Affiliations:** 1Department of Chemistry, University of Turin, 10125 Turin, Italy; maria.tannous@unito.it (M.T.); gjylije.hoti@unito.it (G.H.); 2Department of Drug Science and Technology, University of Turin, 10125 Turin, Italy; roberta.cavalli@unito.it; 3Department of Clinical and Biological Sciences, University of Turin, 10043 Orbassano, Italy; pasquale.pagliaro@unito.it (P.P.); claudia.penna@unito.it (C.P.); 4Hayashibara Co., Ltd., 675-1 Fujisaki, Naka-ku, Okayama 702-8006, Japan; takanobu.higashiyama@hb.nagase.co.jp

**Keywords:** oxygen delivery, cardioplegic solution, α-cyclodextrin, cyclic nigerosyl-nigerose, nanosponges, prolonged release, hypothermia, organ transplantation, organ explantation

## Abstract

Nanocarriers for oxygen delivery have been the focus of extensive research to ameliorate the therapeutic effects of current anti-cancer treatments and in the organ transplant field. In the latter application, the use of oxygenated cardioplegic solution (CS) during cardiac arrest is certainly beneficial, and fully oxygenated crystalloid solutions may be excellent means of myocardial protection, albeit for a limited time. Therefore, to overcome this drawback, oxygenated nanosponges (NSs) that can store and slowly release oxygen over a controlled period have been chosen as nanocarriers to enhance the functionality of cardioplegic solutions. Different components can be used to prepare nanocarrier formulations for saturated oxygen delivery, and these include native α-cyclodextrin (αCD), αcyclodextrin-based nanosponges (αCD-NSs), native cyclic nigerosyl-nigerose (CNN), and cyclic nigerosyl-nigerose-based nanosponges (CNN-NSs). Oxygen release kinetics varied depending on the nanocarrier used, demonstrating higher oxygen release after 24 h for NSs than the native αCD and CNN. CNN-NSs presented the highest oxygen concentration (8.57 mg/L) in the National Institutes of Health (NIH) CS recorded at 37 °C for 12 h. The NSs retained more oxygen at 1.30 g/L than 0.13 g/L. These nanocarriers have considerable versatility and the ability to store oxygen and prolong the amount of time that the heart remains in hypothermic CS. The physicochemical characterization presents a promising oxygen-carrier formulation that can prolong the release of oxygen at low temperatures. This can make the nanocarriers suitable for the storage of hearts during the explant and transport procedure.

## 1. Introduction

The safe capture, storage, and adsorption of large quantities of gas are some of the major technological and scientific challenges facing scientists today. Gas storage is a topic that affects several research fields, as well as climate change, energy production, and medical applications. In particular, oxygen assumes a great therapeutic value in the treatment of solid tumors. The latter is characterized by a highly hypoxic environment that greatly limits current therapies such as chemotherapy, radiotherapy, photodynamic therapy, and even immunotherapy. Different approaches are used to increase the oxygen concentration in tumor tissues. In particular, perfluorocarbons capable of solubilizing large quantities of oxygen are used. However, they cause severe inflammation, release oxygen in an uncontrolled manner, and, above all, are immiscible in both hydrophilic and hydrophobic environments. For these reasons, it is essential to find materials capable of transporting oxygen and releasing it in a controlled way that is biocompatible, non-toxic, and easily synthesized and therefore easily transferable to industrial production [1,2,3,4]. Cyclodextrins (CDs) are natural cyclic oligosaccharides with a lipophilic central cavity and a hydrophilic outer surface consisting of (α-1,4-)-linked α-D-glucopyranose units. They contain six (αCD), seven (βCD), and eight (γCD) D-glucose units. The presence of numerous reactive hydroxyl groups on CDs enables their chemical modification, using a wide variety of bi- or polyfunctional chemicals and thus tailoring their structure. Consequently, water-soluble and insoluble cyclodextrin-based polymers are produced. The insoluble cyclodextrin-based polymers or cyclodextrin-based nanosponges (CD-based NSs) are chemically cross-linked polymers obtained by reacting the parent CDs with an appropriate cross-linking agent such as dianhydrides, active carbonyl compounds, carboxylic acids, epoxides, diisocyanates, etc. CD-based NSs are three-dimensional polymer networks with remarkable adsorption properties due to their extensive nanometer-sized porosity. The most outstanding feature of CD-based NSs is their capability to form inclusion complexes with a wide range of liquid, solid, and gaseous compounds through a molecular complexation. CD-based NSs have shown more benefits than CDs to improve the stability, release, and bioavailability of complex guest molecules. This is because the CDs’ cross-linking agent molar ratio affects the nanochannels produced [5,6,7,8,9,10,11].

Various parameters, such as chemical composition, pore size, and thus surface area and pore volume, can be effectively modulated, even in theoretical simulation models, to optimize the gas adsorption capacity of nanosponges (NSs).

The ability of CD-based NSs to bind compounds reversibly, even in the gas phase, does not require high pressures and low temperatures and thus avoids the hazards of handling high-pressure compressed gases and the technological constraints associated with the low temperatures of liquefied gas. This is possible via physical absorption mechanisms, which may lead to the development of a new technology for the efficient storage of significant amounts of gas in strikingly small volumes [12,13].

CD-based NSs, particularly those prepared based on alpha cyclodextrins (αCD), have proven themselves to be highly suitable for the storage of gases. The ring that makes up αCD resembles a conical cylinder, or truncated cone, in the shape of a crown and includes six glucopyranose units that form a cavity, which is lined by hydrogen atoms that form glycosidic oxygen bridges. The non-bonding electron pairs of these oxygen glycosidic bridges point towards the internal part of the cavity, yielding high electron density and providing it with some basic Lewis properties. Its internally hydrophobic and externally hydrophilic nature means that αCD is principally used for stabilizing reactive intermediates and trapping small molecules (Figure 1) [14,15].

On the other hand, cyclic nigerosyl-1,6-nigerose (CNN), also known as cyclo tetra glucose, is also worth considering for gas storage. It is naturally found in sake (the sediment formed during rice-wine production) and *Saccharomyces cerevisiae* cells. Indeed, CNN is a natural, novel, non-reducing carbohydrate in which four glucopyranose units are bound by alternating α-1,6 and α-1,3 glycosidic linkages.

Numerous potential applications for sugar-based NSs have been investigated in several technological fields, including pharmaceuticals, cosmetics, catalysis, gas storage, agriculture, and polymer additives. One of the most promising areas of investigation is the application of sugar-based NSs as novel drug-delivery systems. The use of oxygenated cardioplegic solution (CS) during cardiac arrest may be beneficial. Indeed, cold, fully oxygenated crystalloid solutions can be an excellent means of myocardial protection [16]. However, the forced oxygenation of a cardioplegic solution, while maximizing its oxygen content, may result in it becoming highly alkaline. The equilibration of a bicarbonate-containing solution with 100 percent oxygen decreases the amount of dissolved carbon dioxide and consequently that of carbonic acid [17].

Altering the pH of a CS can undoubtedly affect the functional recovery of an explanted organ. Furthermore, the concentration of hydrogen ions alters with cooling, as the dissociation constant of any salt solution is a function of temperature. Therefore, when introducing a gas into a cardioplegic solution, it is necessary to control both the temperature of the gas and the temperature at which the solution is delivered, as they interact to determine the final pH and concentration of the gas [18,19].

Oxygenated nanosponges that can store and slowly release oxygen over a controlled time have been chosen as nanocarriers to enhance the functionality of cardioplegic solutions for this purpose. Different formulations are used to prepare nanocarriers for oxygen delivery, and these include native αCD, α-cyclodextrin nanosponges (αCD-NSs), native CNN, and cyclic nigerosyl-nigerose nanosponges (CNN-NSs), among others, which have proven to be innovative tools for controlled and prolonged oxygen delivery.

The addition of nanocarriers with desirable versatility to cold (4 °C) cardioplegic solution, which can be sterilized to reduce the risk of infections, can be naturally decomposed to release more oxygen, thereby improving the functional recovery and/or prolong the amount of time that the heart stays in static hypothermic cardioplegic solution (so-called “static cold storage”, SCS). Thus, it can be theoretically possible to extend the time in hypothermia and to use organs explanted in facilities far away from the recipient’s location, thus exceeding the 4–5 h that have canonically been tolerated so far [20].

Because temperature greatly influences the solubility of a gas in a liquid, the ability of oxygen-loaded formulations to store and slowly release oxygen over time was evaluated at 4 °C, at room temperature (RT, 23 °C), and 37 °C. The chosen temperatures were the ones utilized in laboratory and clinical practice. Specifically, 4 °C mimics the hypothermic situation in which organ metabolism is low (i.e., SCS), while room temperature can be considered the sub-normothermic state [21].

This investigation aimed to compare two diverse biocompatible nanodevices (CNN-NSs and αCD-NSs) as oxygen reservoirs. This study displays for the first time the ability of synthesized nanosponges (CNN-NSs and αCD-NSs) to encapsulate, store, and release oxygen into the cardioplegic solution for a more prolonged period.

These naturally proposed nanocarriers can be sterilized, infused into the organ before explantation, and added to the cardioplegic solution. CNN-NSs presented as a more controlled oxygen release than CD-NSs. These findings will serve such a perspective for the further investigation of indicated nanodevices with pre-clinical studies and thereafter clinically as cardioprotective agents.

## 2. Results

The oxygenation of a cardioplegic solution (CS) can likely improve explanted heart vitality and functions.

Several strategies have been studied to increase the concentration of CS oxygen from red blood cells to artificial oxygen carriers, such as hemoglobin-based oxygen carriers, extracellular vesicles, per-fluorocarbon emulsions, and nanoparticulate systems.

In the present work, two glucose derivatives, αCD and CNN, and two cross-linked polymers between them that form nanoparticles called nanosponges (NS) have been investigated in an assessment of their oxygen-carrier capacity in the National Institutes of Health (NIH) CS. Both αCD and CNN are cyclic oligosaccharides with a central cavity (see Figure 1) that can encapsulate molecules, including gases, which can be entrapped and stored in the α-CD and CNN cavities and the nanoporous matrices of αCD and CNN nanosponges, as previously demonstrated. Oxygen-loaded αCD- and CNN-based nanosponges have recently been demonstrated to protect against hypoxia/reperfusion (H/R)-induced cell death in cell experiments [22,23].

In this work, the effect of the addition of oxygenated αCD, CNN, αCD-NS, and CNN-NS on the oxygenation of NIH CS (National Institutes of Health Cardioplegic Solution) was evaluated at different temperatures and concentrations to increase and prolong the oxygen content over time.

First, the αCD- and CNN-based nanosponges were successfully synthesized according to a previously tuned procedure, and their stability in NIH CS was evaluated.

Four oxygenated formulations were then prepared with the addition of αCD, CNN, αCD-NS, and CNN-NS to NIH CS and loaded with oxygen. The formulations were characterized, and their physicochemical parameters (Table 1) were determined.

The basic pH value of the cardioplegic solution is due to the presence of bicarbonate in its composition. The presence of the two compounds slightly decreased the pH of NIH CS. Interestingly, the addition of O_2_ maintained the basic pH value. The viscosity increased only in the presence of CNN oligosaccharide, but the solution is still suitable for organ perfusion.

Table 2 reports the physicochemical characteristics of NIH CS in the presence of the two types of oxygen-loaded nanosponges. The two formulations are nanosuspensions, which, from the physical point of view, are αCD- and CNN-based nanosponges and consequently solid water-insoluble nanoparticles.

Both the oxygen-loaded CNN-NSs and αCD-NSs were around 500 nm in size and had a negative surface charge, with zeta potential values high enough to avoid aggregation phenomena in NIH CS.

The sizes and surface charges of the two NSs formulations in NIH CS are similar to those measured in water. Only the PDI showed a slight increase, and this can be correlated to the salts present in the solution.

The addition of oxygen-loaded NSs did not increase the pH value, as the oxygen is stored in the nanocarriers and not immediately available in the solution. This behavior can overcome the limitations of the formation of more alkaline solutions because of the low buffering capacity of the bicarbonate concentration present.

The viscosity-value increase in NIH CS was negligible in the presence of αCD-NSs, while it reached a value of 1.56 mPAs with the addition of CNN-NSs. Nevertheless, the two values are suitable for the potential clinical application of the formulations.

The production of Reactive Oxygen Species (ROS), which can induce cell damage and apoptosis, is a limitation in oxygen delivery. The amount of ROS in oxygenated NIH CS formulations was determined and compared with formulations that are not saturated with oxygen. The results are reported in Table 3. Interestingly, the addition of αCD and αCD-NSs did not significantly increase the ROS concentration compared to the value of NIH CS.

Because temperature remarkably influences the solubility of a gas in a liquid, the capability of the oxygen-loaded formulations to store and slowly release oxygen over time was evaluated at 4 °C, at room temperature (RT, 23 °C), and 37 °C.

Figure 2 reports the oxygen profiles of gas diffusion from the oxygenated NIH CS at three different temperatures for up to 24 h.

Slight differences in oxygen content were observed between room temperature and 37 °C (Table 4) after the first hours.

Indeed, as the temperature increases (4 °C, room temperature (RT), and 37 °C), the solubility of a gas decreases from 29.60 mg/L to 18.64 mg/L because of the tendency of a gas to expand [24,25]. Moreover, the presence of salts in the NIH CS can play a crucial role in favoring the liberation of gas from solution.

Henry’s law states that “at a constant temperature, the amount of a given gas that dissolves in a given type and volume of liquid is directly proportional to the partial pressure of that gas in equilibrium with that liquid”.

The oxygen concentration in NIH CS depends on Henry’s law and, after 24 h, the amount of oxygen reaches equilibrium with the atmospheric content.

The oxygen-saturated NIH CS was considered as a reference in testing the ability of the oxygen-carrier-containing formulations to store and prolong the release kinetics of oxygen over time.

Figure 3 reports the kinetics of oxygen release from oxygenated NIH CS that contained CNN, which was added at a concentration of 1.3 g/L, at different temperatures. The graph also confirms the effect of temperature on the amount of dissolved oxygen in this NIH CS formulation, although the presence of CNN markedly affects the amount of dissolved oxygen in the CS solution and produced sustained release over time. Specifically, the oxygen-carrier capability of CNN is evident at 4 °C.

The addition of CNN played an important role in sustaining oxygen release for a longer time in the NIH CS solution, as it can encapsulate oxygen in the inner cavity of the molecule.

The prolonged oxygen release is demonstrated by the greater O_2_ concentration at 4 °C, which can be compared to the marked gas-concentration drop in NIH CS after 6 h. Moreover, the oxygen concentration is still higher after 24 h. This behavior may be promising for the enhancement of heart storage before transplant in static cold storage [17].

Interestingly, the capability of CNN to sustain oxygen release was enhanced with the addition of CNN-NSs in NIH CS.

The benefits that the CNN-NSs display over the CNN oligosaccharide in the NIH CS are underlined by a slower and more controlled release achieved using the nanosponges, at least over the first 6 h of the recorded oxygen release. This is in contrast to the quick decrease in oxygen concentration in the CNN-containing cardioplegic solution. The entrapment of oxygen is also evident at 37 °C (Figure 4), and this behavior underscores the crucial role played by the polymer network in the matrix and by the CNN cavities, as reported in Figure 5.

The capability of αCD to act as an oxygen nano reservoir is also demonstrated.

The kinetics of oxygen release from αCD in CS at different temperatures is shown in Figure 6.

Similar to the CNN results, the αCD, dissolved at a concentration of 1.3 g/L in NIH CS and loaded with oxygen, increased oxygen concentration in the solution. Moreover, αCD displayed faster release at higher temperatures compared to the αCD-NSs nanoformulations, as presented in Figure 7.

The kinetics of oxygen release from the oxygen-loaded αCD-NSs displays a marked difference compared to results with αCD oligosaccharide. Oxygen encapsulation was greater in the nanosponges due to the presence of the polymer network caused by the presence of a cross-linking agent in the NSs matrix and the cooperation of the αCD cavities (Figure 8), as previously observed with CNN-NSs.

CS released a higher concentration of oxygen over the first six hours (Figure 8). However, 24 h later, the αCD-NSs had a higher oxygen concentration than that of αCD alone or CS.

The results showed that oxygen-release kinetics were slower for CNN in CS than that for αCD since CNN is a tetraglucose that has a smaller cavity and is more polar than αCD, which consists of six glucose units; the smaller cavity of CNN favors oxygen encapsulation.

The effect of oxygen-loaded nanosponge concentration on the gas storage and delivery capability was then investigated (Table 5). The NSs retained more oxygen at 1.3 g/L than when diluted to 0.13 g/L and showed no significant increase at a concentration of 13.0 g/L, with this effect being more marked at 37 °C.

This behavior suggests that small amounts of nanosponge can be used to obtain a suitable oxygen concentration.

The histograms (Figure 9, Figure 10, Figure 11, Figure 12 and Figure 13) highlight the oxygen percentages recorded at a certain time over the 24 h release study. CNN-NSs and αCD-NSs are more adequate nanocarriers than the parent CNN and αCD, thus validating the concept of storing oxygen in the cardioplegic solution for at least 12 h after the loading procedure. The tuning of the flux time, concentration, storage conditions, and controlled environment are areas for further study and can strengthen the rationale and support the results.

## 3. Discussion

Nanocarriers as a tool to improve the delivery and storage of oxygen have been the focus of extensive research in the field of organ transplants. The static cold storage (SCS) of the donor’s heart after brain death remains the clinical standard. The main advantage of hypothermia is to “prevent” ischemic damage when the cardioplegic flow is interrupted. This advantage is achieved by a reduction in the temperature-dependent rate of myocardial metabolism, which results in a slowing of the rate of ischemic damage. However, even when drastically reduced by cold cardioplegia, the heart continues to have a minimally oxygen-demanding metabolism [26,27].

The automated machine perfusion (MP) of a donor’s heart is being evaluated as an alternative approach to donor-organ management and as a means to expand the donor pool and/or increase the utilization rate. However, technical and biological issues (e.g., machine malfunction, user error, infections, and costs) limit its large-scale use, and further well-designed studies are needed to draw clear conclusions.

Previous studies have demonstrated the ability of αCD-NSs and CNN-NSs to accommodate and convey oxygen, thus addressing growth problems in several diseases, from inflammation to cancer, and improving tumor treatment. The αCD-based formulations can be used for the treatment of cardiovascular diseases as oxygen nanocarriers that can limit ischemic reperfusion injury (IRI) via direct injection into the myocardial wall before starting full-blood reperfusion. Similarly, the proposed CNN-based nanocarriers have shown marked efficacy in controlled oxygenation and have effectively protected cellular models (e.g., cardiomyocytes and endothelial cells) from simulated IRI, thus reducing cell mortality [13,22,23,28,29]. The polymerization reactions lead to a successful synthesis of carbonate NSs. The resulting products were further investigated to understand the difference between CNN and α-CD as building blocks. When the CNN and α-CD react with CDI, two OH groups are completely esterified, and the reaction results in imidazole formation. The cross-linking process is confirmed via carbonate formation. The formed product is more resistant to hydrolysis than, for example, the synthesized-based NSs based on carboxylic acid esters [8].

In this novel study, two different biocompatible low-temperature oxygen-releasing nanodevices (CNN-NSs and αCD-NSs) that can be easily sterilized, infused into the organ before explantation and added to the cardioplegic solution under SCS conditions, where they release oxygen over a long period, are evaluated. Under these conditions, the organs to be transplanted can be theoretically kept under oxygenated conditions in SCS for a longer time, with the belief that these nanodevices can be first verified experimentally with pre-clinical studies and later clinically as cardioprotective agents.

The two kinds of nanocarriers were capable of playing the relevant roles of oxygen reservoirs in this evaluation of their oxygen-encapsulation capability. The two NSs were compared with their building units, i.e., αCD and CNN. Interestingly, the NSs nanostructure can affect oxygen storage and release, and CNN-NSs have been demonstrated to be a better oxygen reservoir than CD-NSs.

This behavior may be related to the different nanostructures of the two nanocarriers. Compared with CNN, αCD has a much larger and more hydrophobic cavity. CNN has two inward-oriented hydroxyl moieties, so only small molecules can be included within them. Nevertheless, similarly to CD, CNN-NSs have two types of nanopores: the interstitial spaces between the CD units and the internal hydrophobic CD cavities. The spaces between the CD can be more or less hydrophilic, depending on cross-link density and cross-linking agent polarity. Due to a large number of reactive hydroxyl groups, starch derivatives can act as polyfunctional monomers and be cross-linked using a wide array of chemicals (e.g., active carbonyl compounds, diisocyanates, dianhydrides, epoxides, carboxylic acids with two or more functionalities, etc.), thus yielding three-dimensional, insoluble polymers [8], with specific characteristics and capability for molecule encapsulation.

As it is known, the formation of an inclusion complex involves partial or complete coverage of the host molecule. This provides, on the one hand, protection from evaporative degradation and oxidation and allows for the stabilization of the host, while on the other, it alters many of the physicochemical properties of the molecule. These modifications facilitate the use of characterization techniques to verify that the host is indeed contained in the cavity. αCD and CNN may maintain some of their ability to form hydrogen bonds with other molecules, allowing them to stay, at least partially, soluble in water in the form of an inclusion complex [30]. Furthermore, αCD and CNN cross-linking insoluble nanodevices are obtained with increased oxygen encapsulation capacity.

In addition, the storage of oxygen in nanodevices and the subsequent slow and constant delivery avoid the formation of high ROS species that can damage heart cells in the NIH CS. It is a key point to the limitation of free radical species in transplantation to maintain organ function.

Oxygen-release kinetics at low temperatures varied according to the nanocarrier used, suggesting that they may be appropriate and tuned for typical SCS temperatures (4 °C). Nevertheless, this aspect requires verification via the use of a transplanted heart and clinical applications.

Unquestionably, either a slight or severe oxygen deficiency can lead to cell death, but a wealth of research suggests that fluctuating oxygen levels rather than persistently low pO2 are the most harmful aspects [31]. The most intriguing aspect of oxygen consumption by the heart is that it can switch from low to high consumption levels in response to metabolic demand. As a result, we can suggest that the intracellular compartment experiences a range of oxygen tensions, from high to low, depending on the balance between supply and demand. A pro-oxidative environment is created by a positive balance. In this context, the changes in oxygen levels (dysoxia) switching from hypothermia to normothermia can be crucial. Indeed, interactions of oxygen with other gases, such as nitric oxide, can either aggravate or protect ischemic organs, depending on several circumstances and including the relative level of the two gases. Additionally, the effects of dysoxia are crucial for iron metabolism. Therefore, only future studies can reveal the real effects of controlled oxygen delivery by nanodevices into transplanted organs that are first stored at low temperatures and then transplanted into a warm body.

## 4. Materials and Methods

### 4.1. Chemicals

α-Cyclodextrin (α-CD, M_w_ = 972.846 g/mol) was a gift from Roquette Frères SA (Lestrem, France). Cyclic nigerosyl-1-6-nigerose (CNN, M_w_ = 648.564 g/mol) was obtained from Hayashibara (Tokyo, Japan). CNN and α-CD were desiccated in an oven at 80 °C up to constant weight before their usage to remove any traces of absorbed moisture. Additionally, 1,1′-Carbonyldiimidazole (CDI, ≥97.0%), 2′,7′-dichlorodihydrofluorescein diacetate (H2DCFDA fluorescent probe), and 2′,7′-dichlorofluorescein (DCF), N,N-dimethylformamide (DMF, ≥99.8%), acetone (≥99% (GC)), and ethanol (96.0–97.2%) were purchased from Sigma-Aldrich (Munich, Germany). Deionized and MilliQ^®^ water was obtained using a Millipore Direct-QTM 5 production system. All other chemicals used to prepare the cardioplegic solution (CS) were commercially available as analytical-grade products.

### 4.2. CNN-Based Nanosponge Synthesis

CNN-based nanosponges (CNN-NSs) were successfully synthesized following an existing procedure in literature with minor modifications [22]. The synthesis was performed by dissolving 5.00 g (7.70 mmol) of anhydrous CNN (Figure 1) powder in 30 mL DMF at room temperature in a round bottom flask, using a hotplate stirrer equipped with thermoregulation and heat-on block. Subsequently, 4.99 g (30.70 mmol) of CDI as a cross-linking agent was added, and to observe a clear solution, the temperature was increased up to 80 °C until the gel formation. Additionally, the formed gel was kept at 90 °C for around 5 h until a solid product was obtained. The stoichiometric molar ratio between CNN and CDI was 1:4. The acquired monolithic block was then broken up and manually ground in a mortar. The product was further purified with an excess of deionized water and recovered using a Buchner filtration system using filter paper (Whatman No. 1, Whatman, Maidstone, UK). The by-products were completely removed through Soxhlet extraction using ethanol for around 24 h. Finally, the CNN-NSs were air-dried, milled, and exploited for characterization as a white homogeneous powder.

### 4.3. αCD-Based Nanosponge Synthesis

The same synthesis, purification, and recovery procedures, as previously described for CNN-NSs, were utilized to obtain αCD-based nanosponges (αCD-NSs). Briefly, 5.00 g (5.14 mmol) of αCD and 3.33 g (20.54 mmol) of CDI were dissolved in 30 mL DMF. The stoichiometric molar ratio between α-CD and CDI was 1:4. A schematic representation of the synthesis of αCD-NSs is shown in Figure 14.

### 4.4. Preparation of Cardioplegic Solution

The National Institutes of Health (NIH) cardioplegic solution (CS) used throughout our experiments was freshly prepared according to the National Institute of Health for Cardioplegic Protection protocol. The formulation is reported in Table 6 with a slight modification [32].

### 4.5. Preparation of Nanoformulations

The defined amounts of CNN and αCD powders were weighed and dissolved in the prepared NIH cardioplegic solution to obtain solutions with different concentrations. A weighed amount of finely milled NSs powder was first added to the cardioplegic solution and then homogenized with a high-shear mixer (Ultra-Turrax^®^, Konigswinter, Germany) at 24,000 rpm for 10 min to reduce the size and obtain well-distributed suspensions with a uniform particle size. The aqueous suspension was later subjected to an ultrasonic sonicator for 20 min to achieve homogenization. The obtained nanosuspensions were then purified and stored in the fridge. All nanoformulations were stable after one week at 4 °C.

### 4.6. Characterization of Nanoformulations

The different NSs formulations were further characterized in vitro to evaluate their size and surface charge. The average diameter and polydispersity index of the formulations were measured using photon correlation spectroscopy (PCS) with a 90 Plus instrument (Brookhaven, NY, USA) at a fixed angle of 90 and a temperature of 25 °C after dilution with filtered water. The zeta potential was determined using a 90 Plus instrument (Brookhaven, NY, USA). For zeta-potential determination, diluted samples of nanoformulations were placed in an electrophoretic cell, to which a rounded 15 V/cm electric field was applied. The pH was recorded at room temperature using an Orion 420A pH meter.

### 4.7. Oxygen-Loading Procedure

To prepare oxygen-loaded nanoformulations, different quantities of αCD, CNN, αCD-NSs, and CNN-NSs were weighed and placed into a vial with the prepared NIH cardioplegic solution. After shear homogenization and ultrasound sonication, NSs dispersions were transferred into a three-neck round-bottom flask and saturated with an oxygen purge at a flux of 4 L/min under stirring, while the gas concentration was monitored up to 35 mg/L in the external aqueous phase. The stability of the oxygen-encapsulating NSs was evaluated over time at different temperatures by measuring the oxygen concentration inside sealed falcon tubes using the oximeter electrode mounted inside.

### 4.8. Oxygen-Release Profile

The dissolved oxygen (DO) concentration was recorded using an oxygen bench meter (HI5421, Hanna Instruments Inc., Woonsocket, RI, USA). The bench meter is supplied with a probe for laboratory use and a built-in temperature sensor (HI76483 Hanna Instruments Inc. Woonsocket, RI, USA) was inserted inside a sealed falcon tube. This allowed the measurements of DO to be carried out in a closed system and thus avoided the interferences from the oxygen in the environment. A fixed volume of 45 mL of oxygen-loaded NIH CS solution/nanosuspension was purged with oxygen flux for 30 min. After a predetermined incubation and equilibration time, the cap of the falcon tube was removed and replaced with the electrode for DO measurement. The oxygen concentration was logged every 30 min for 24 h. To minimize fluctuations in the readings caused by environmental and instrumentation factors, both a thermostat and a cryo-compact circulator (JULABO GmbH, Seelbach, Germany) were used to regulate the temperature (4–37 °C) of the medium in the falcon containing the solution and the probe. All measurements were carried out in triplicate under determined conditions.

### 4.9. Determination of Reactive Oxygen Species (ROS)

The fluorescence method was used, based on the oxidation of the H2DCFDA fluorescent probe (2′,7′-dichlorodihydrofluorescein diacetate), for the determination of the oxygen free radicals present in the cardioplegic-solution samples.

For this purpose, a 2 mL ethanolic solution of H2DCFDA (10 mM) was added to 0.5 mL of 0.01 M NaOH to hydrolyze H2DCFDA into DCFH (non-fluorescent compound). The hydrolysis product was maintained at room temperature for 30 min and neutralized by adding 10 mL of PBS (50 mM, pH 7.2). The presence of ROS in the DCFH compound is rapidly oxidized to 2′,7′-dichlorofluorescein (DCF).

The green fluorescence color of DCF was measured using a fluorimeter (EnSightTM automated multimode plate reader, PerkinElmer, Inc., Waltham, MA 02451, USA) with the excitation wavelength set at 485 nm and emission at 530 nm. The concentration of ROS was determined using a calibration curve built by analyzing a series of standard solutions of DCF in the 0.001–0.500 µM concentration range. Standard solutions were prepared by diluting a stock solution of DCF (500 µM) with phosphate buffer saline solution (PBS).

A linear calibration curve was obtained in the concentration range between 0.001 and 0.500 µM, with an R^2^ value of 0.998.

### 4.10. Quantitative Determination of ROS in Blank/Oxygen-Loaded Samples

The safety profiles of the αCD-NSs and CNN-NSs have been thoroughly investigated as candidates for several pharmaceutical applications [33,34,35,36]. Their cytotoxicity on anaplastic thyroid cancer cells and internalized in the A2780 cell line have been evaluated in previous studies [28,37].

For the determination of ROS, the blank cardioplegic solution CS, the prepared solutions, and the nanosuspensions with the αCD, CNN, αCD-NSs, and CNN-NSs (concentration 10 mg/mL) were loaded with oxygen for 30 min before being tested. In separate vials, 2 mL of each sample was pipetted, and 25 μL of the prepared DCFH was added to reach a final concentration of 5 µM. The samples were left at room temperature and protected from light for 20 min to achieve reaction completion. The fluorescence intensity was then measured with a fluorimeter (excitation 485 nm, emission 530 nm) for 60 min. The analyses were performed in triplicate on each sample.

## 5. Conclusions

In this study, two oligosaccharides, i.e., CNN and αCD, and their cross-linked polymers, namely CNN-NSs, and αCD-NSs were extensively studied as artificial oxygen nanocarriers. Oxygen-release kinetics were slower for CNN than for αCD, and the results showed that the polymer network can play a key role in oxygen storage and release. CNN-NSs and αCD-NSs display the ability to encapsulate, store, and release oxygen into the cardioplegic solution for a more prolonged period.

The proposed nanocarriers have considerable versatility; they can be sterilized by their addition to the cardioplegic solution which reduces the risk of infections, and they can be naturally decomposed so that they release oxygen. This leads to potentially prolonging the amount of time that the heart stays in the hypothermic cardioplegic solution. Furthermore, the nanocarriers can be used for distant organ transport, thus exceeding the current canonically tolerated time (around 6 h). These findings are promising for the tuning of an oxygen-carrier formulation that can prolong the storage of the heart during the explantation and transport procedures.

## Figures and Tables

**Figure 1 ijms-24-10073-f001:**
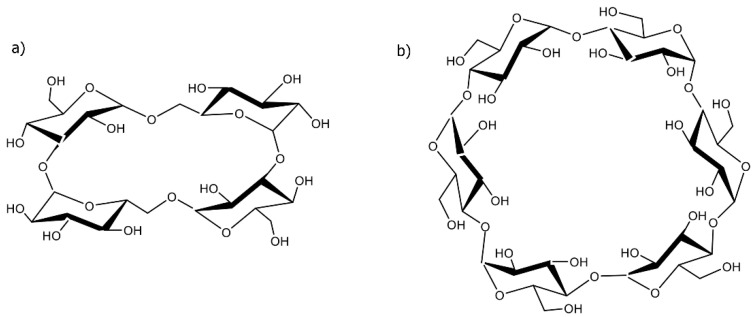
Chemical structures of (**a**) cyclic nigerosyl-1,6-nigerose (CNN) and (**b**) α-cyclodextrin (αCD).

**Figure 2 ijms-24-10073-f002:**
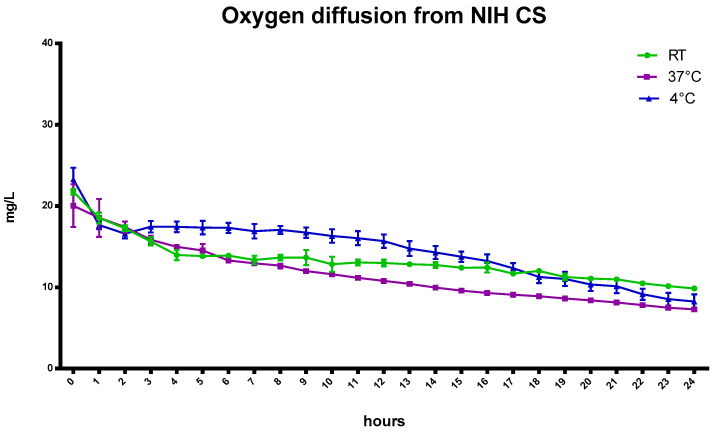
Oxygen diffusion from NIH-oxygen-saturated cardioplegic solutions at different temperatures.

**Figure 3 ijms-24-10073-f003:**
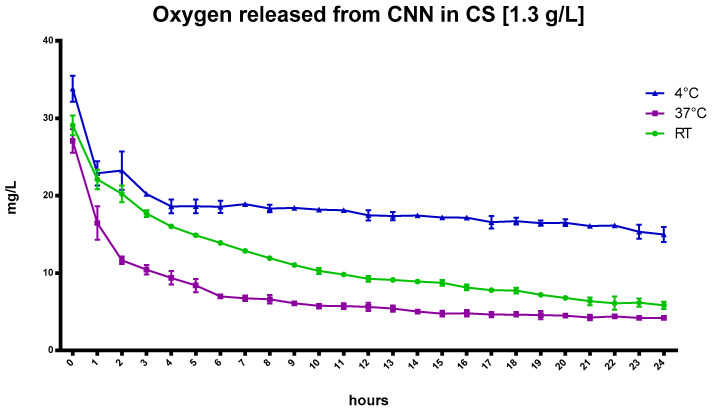
In-vitro release kinetics of oxygen from CNN in cardioplegic solution CS at a concentration of 1.3 g/L at different temperatures.

**Figure 4 ijms-24-10073-f004:**
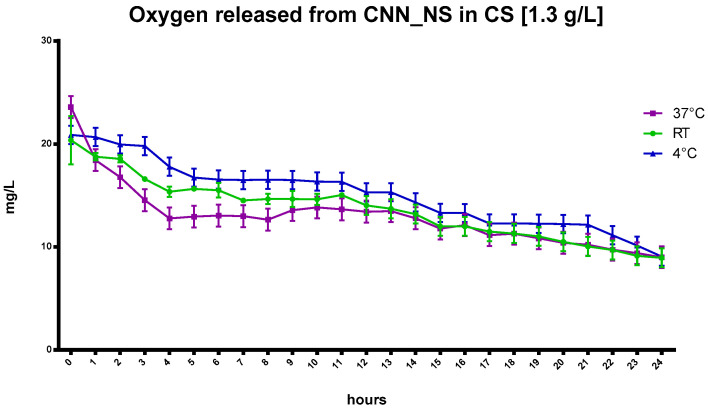
The kinetics of in-vitro oxygen release from saturated cardioplegic solution with CNN-NSs at a concentration of 1.3 g/L at different temperatures.

**Figure 5 ijms-24-10073-f005:**
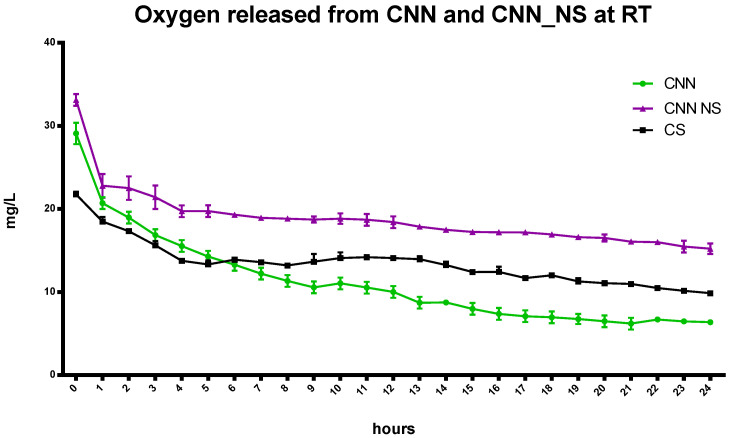
Comparison of the kinetics of the in-vitro release of oxygen from CNN and CNN-NSs in CS at the same concentration at room temperature.

**Figure 6 ijms-24-10073-f006:**
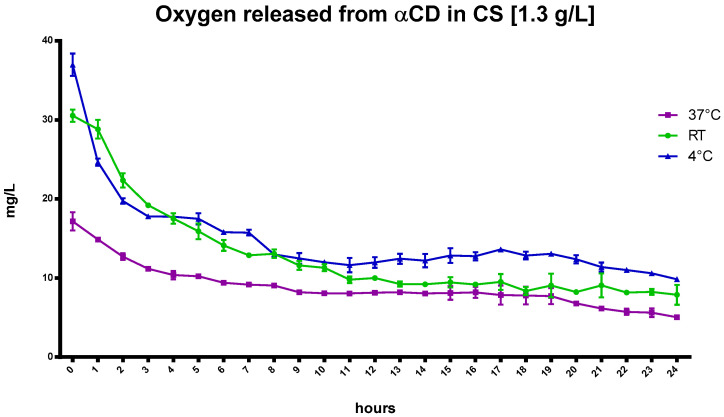
The kinetics of in-vitro oxygen release from saturated CS with αCD oligosaccharide at a concentration of 1.3 g/L at different temperatures.

**Figure 7 ijms-24-10073-f007:**
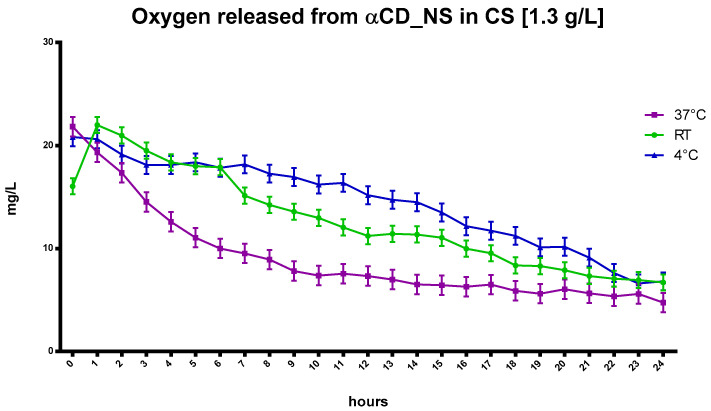
The kinetics of in-vitro oxygen release from saturated CS with αCD-NS at a concentration of 1.3 g/L at different temperatures.

**Figure 8 ijms-24-10073-f008:**
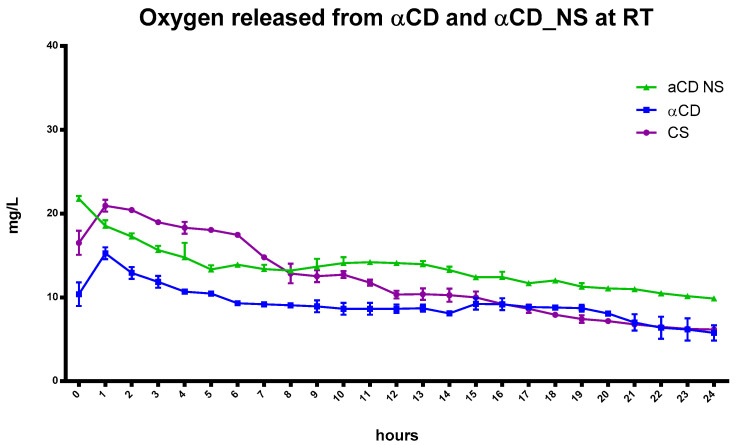
Comparison of the in-vitro release kinetics of oxygen from αCD, αCD-NSs, and CS at room temperature.

**Figure 9 ijms-24-10073-f009:**
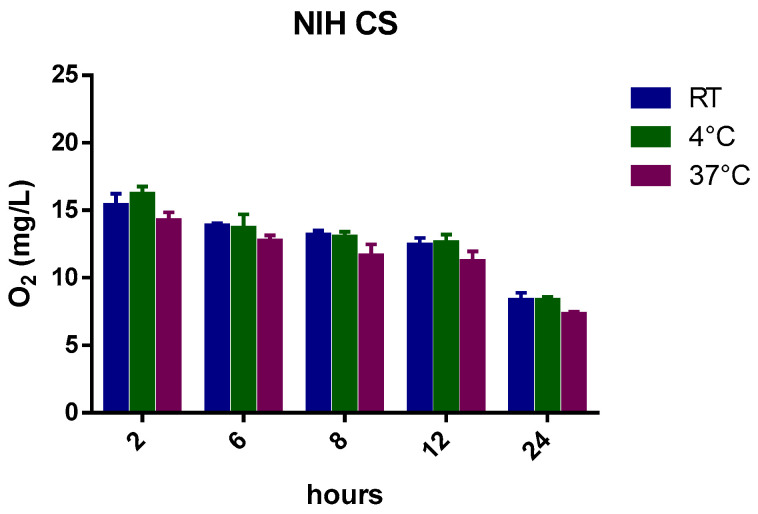
Histogram showing the variation of oxygen concentration in cardioplegic solution at different temperatures.

**Figure 10 ijms-24-10073-f010:**
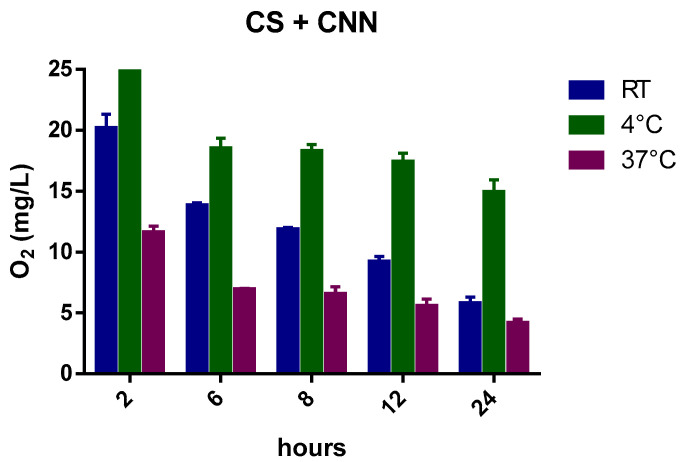
Histogram showing the variation of oxygen concentration in cardioplegic solution + CNN monomer (at a concentration of 1.3 g/L) at different temperatures.

**Figure 11 ijms-24-10073-f011:**
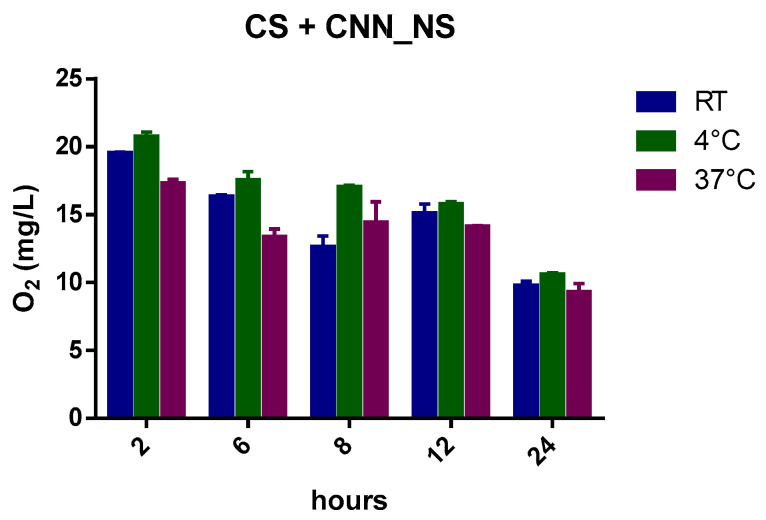
Histogram showing the variation of oxygen concentration in cardioplegic solution + CNN-NSs (at a concentration of 1.3 g/L) at different temperatures.

**Figure 12 ijms-24-10073-f012:**
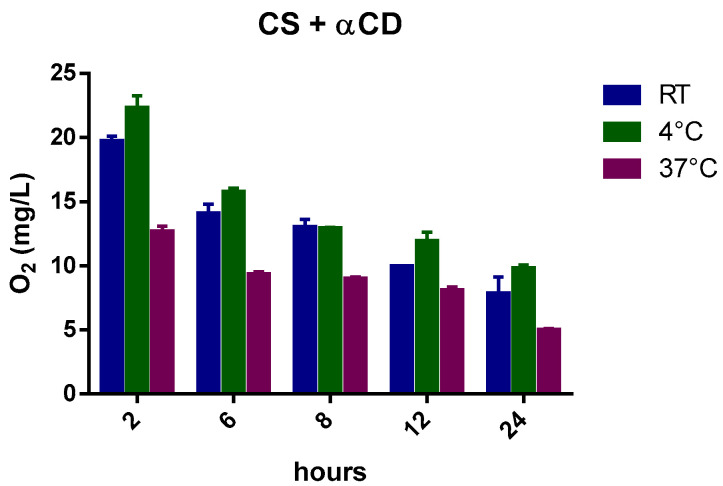
Histogram showing the variation of oxygen concentration in cardioplegic solution + αCD monomer (at a concentration of 1.3 g/L) at different temperatures.

**Figure 13 ijms-24-10073-f013:**
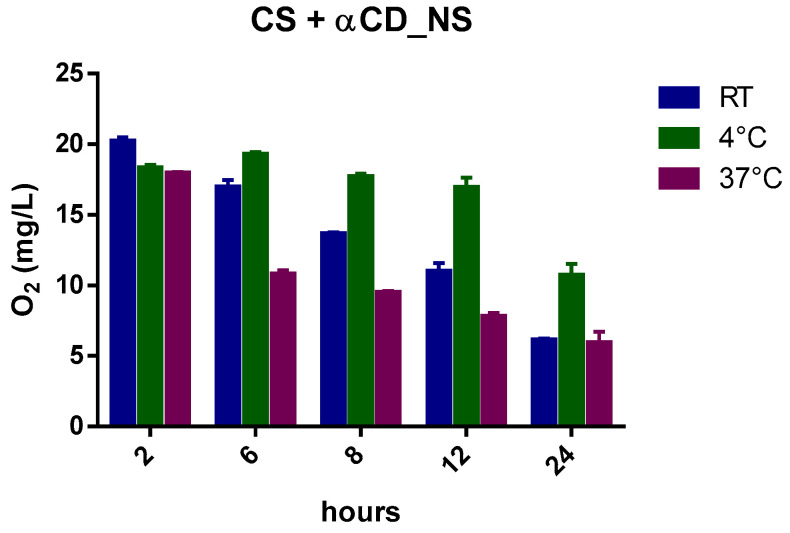
Histogram showing the variation of oxygen concentration in cardioplegic solution + αCD-NSs (at a concentration of 1.3 g/L) at different temperatures.

**Figure 14 ijms-24-10073-f014:**
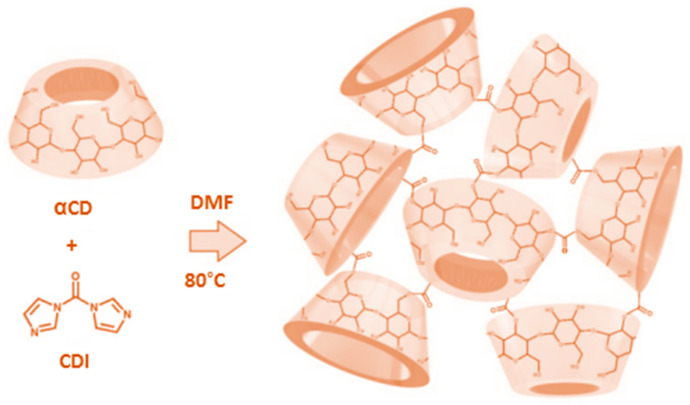
The schematic representation of the reaction of αCD with CDI at certain conditions.

**Table 1 ijms-24-10073-t001:** Physicochemical characterization of NIH CS with αCD and CNN loaded with oxygen.

Sample	NIH CS	NIH CS + CNN	NIH CS + αCD
pH	8.68	8.40	8.21
Viscosity (mPa)	1.20	1.60	1.20

**Table 2 ijms-24-10073-t002:** Physicochemical characteristics of NIH CS with the two types of oxygen-loaded NSs.

Sample	NIH CS + CNN NS	NIH CS + αCD NS
pH	8.50	8.30
Average Diameter ± SD (nm)	485.80 ± 44.10	560.70 ± 35.60
PDI ± SD	0.38 ± 0.01	0.31 ± 0.03
Viscosity (mPa)	1.56	1.26
Zeta Potential ± SD (mV)	−25.58 ± 2.60	−19.30 ± 1.13

**Table 3 ijms-24-10073-t003:** Reactive Oxygen Species (ROS) concentration in the oxygenated NIH CS formulations.

Sample	Concentration (μM)
Water	0.05
PBS	0.01
NIH CS	0.01
CS + O_2_	0.03
αCD in CS	0.04
αCD NS in CS	0.02
αCD in CS + O_2_	0.01
αCD NS in CS + O_2_	0.02
CNN in CS	0.08
CNN NS in CS	0.04
CNN in CS + O_2_	0.01
CNN NS in CS + O_2_	0.03

**Table 4 ijms-24-10073-t004:** The concentration of oxygen in oxygenated NIH CS was recorded after 30 min at different temperatures.

Oxygen Concentration in NIH Cardioplegic Solution	4 °C	RT	37 °C
(mg/L)	29.60 ± 3.01	20.38 ± 2.50	18.64 ± 2.42

**Table 5 ijms-24-10073-t005:** Oxygen concentration in NIH CS with increasing nanocarrier concentration, was recorded at 37 °C for 12 h.

The Concentration of Prepared Nanosuspension (g/L)	The Concentration of Oxygen (mg/L)		
	0.13	1.30	13.0
αCD	5.39 ± 0.31	6.01 ± 0.50	6.12 ± 1.50
CNN	5.63 ± 0.25	5.25 ± 0.32	6.81 ± 1.02
αCD-NSs	6.02 ± 1.03	8.63 ± 1.23	7.38 ± 0.43
CNN-NSs	8.57 ± 1.51	14.52 ± 2.43	11.33 ± 1.73

**Table 6 ijms-24-10073-t006:** Composition of NIH cardioplegic solution.

NIH CS Components	mmol/L
Na	98.9
Cl	107.8
K	30.0
Ca	1.0
HCO_3_	22.0
Glucose	152.6
Mannitol	68.6
Lidocaine, mg/L *	20.0

*—Lidocaine is reported in mg/L.

## Data Availability

The authors confirm that the data supporting the findings of this study are available within the article.

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
