# Peer review of "Oxygen Nanocarriers for Improving Cardioplegic Solution Performance: Physico-Chemical Characterization"

_ijms, 2023, doi:10.3390/ijms241210073_

Round 1
Reviewer 1 Report
The manuscript of some studies on the Oxygen Nanocarriers for Improving Cardioplegic-Solution Performance: physico-chemical characterization.
However, the manuscript has some lacks. Moreover, the study should be given in schematic illustration. Because the study is a little hard to imagine.
1. What is the rationale of study?
2. Language is not proper somewhere. Some grammatical mistakes are present and sentences are not synchronized properly, thoroughly check the whole manuscript.
3. Keywords: Please choose the appropriate and related words in this part.
4. Detailed information about the source, purity, and purification if was carried out of chemicals must be included.
Author Response
Manuscript ID: ijms-2417941
Dear Editor,
thank you for allowing us to submit a revised draft of the manuscript “Oxygen Nanocarriers for Improving Cardioplegic-Solution Performance: Physico-Chemical Characterization”. We appreciate the insightful feedback on our manuscript provided by the reviewers. We would like to thank the reviewers for the careful reading of this manuscript and for the thoughtful comments and constructive suggestions that will lead to valuable improvements in our paper. We have incorporated the suggestions made by the reviewers. They are highlighted within the manuscript by using “Track Changes” (red color) and below, please find the point-by-point responses to the comments and requests of the reviewers. The responses are presented in red.
Reviewer 1- Open Review
However, the manuscript has some lacks. Moreover, the study should be given in schematic illustration. Because the study is a little hard to imagine.
- What is the rationale of study?
Thank you for the question! This investigation aimed to compare two diverse biocompatible nanodevices (CNN-NSs and αCD-NSs) as oxygen reservoirs. This study displays for the first time the ability of synthesized nanosponges (CNN-NSs and αCD-NSs) to encapsulate, store, and release oxygen into the cardioplegic solution for a more prolonged period.
- Language is not proper somewhere. Some grammatical mistakes are present and sentences are not synchronized properly, thoroughly check the whole manuscript.
Thank you very much for the observation. The English language is entirely check in the whole manuscript.
- Keywords: Please choose the appropriate and related words in this part.
Thank you! The keywords are modified.
- Detailed information about the source, purity, and purification if was carried out of chemicals must be included.
Thank you very much for the interesting question. The purification processes carried out for the synthesized nanosponges are added to the synthesis procedures.

Reviewer 2 Report
This manuscript deals with "Oxygen Nanocarriers for Improving Cardioplegic-Solution Performance: physico-chemical characterization." I suggest a minor correction and require a detailed clarification. A correction should be addressed by the authors as follows: The abstract is not well organized; the sentences are incomplete, and there is no sense of continuity. It would be feasible if you included the significance of the current study in the abstract. A brief description of how the authors selected information from the literature in the databases, as well as what time period they searched for, is missing. The authors should justify and expand the information on the advantages of Oxygen Nanocarriers for biomedical applications. Authors should specify the main experimental conditions used based on the evidence from the literature. Where they briefly describe the most important data reported in the literature in a homogeneous manner and reinforce the relevance of Oxygen Nanocarriers as novel alternatives. Authors should discuss whether the use of Oxygen Nanocarriers represents a solid alternative to existing therapeutics. Also, please discuss the use of method using green nanomaterials to targeting cells and mitochondria . Please add the below studies to your manuscript in the discussion section and bold your study novelties:
-Ahmadov, I. S., et al. "The synthesis of the silver nanodrugs in the medicinal plant Baikal skullcap (Scutellaria Baicalensis Georgi) and their antioxidant, antibacterial activity." Advances in Biology & Earth Sciences 5.2 (2020): 103-118.
-Zhang, Junjie, et al. "Nanotechnological strategies to increase oxygen content of the tumor." Frontiers in Pharmacology 14 (2023): 646.
Author Response
Manuscript ID: ijms-2417941
Dear Editor,
thank you for allowing us to submit a revised draft of the manuscript “Oxygen Nanocarriers for Improving Cardioplegic-Solution Performance: Physico-Chemical Characterization”. We appreciate the insightful feedback on our manuscript provided by the reviewers. We would like to thank the reviewers for the careful reading of this manuscript and for the thoughtful comments and constructive suggestions that will lead to valuable improvements in our paper. We have incorporated the suggestions made by the reviewers. They are highlighted within the manuscript by using “Track Changes” (red color) and below, please find the point-by-point responses to the comments and requests of the reviewers. The responses are presented in red.
Reviewer 2 Open Review
This manuscript deals with "Oxygen Nanocarriers for Improving Cardioplegic-Solution Performance: physico-chemical characterization." I suggest a minor correction and require a detailed clarification. A correction should be addressed by the authors as follows:
The abstract is not well organized; the sentences are incomplete, and there is no sense of continuity. It would be feasible if you included the significance of the current study in the abstract.
Thank you very much for the comment! The authors have modified the abstract.
A brief description of how the authors selected information from the literature in the databases, as well as what time period they searched for, is missing.
The authors should justify and expand the information on the advantages of Oxygen Nanocarriers for biomedical applications.
Authors should specify the main experimental conditions used based on the evidence from the literature.
Thank you for the observation! We have added the reference in the synthesis procedure, and modified the experimental procedures of the synthesized nanosponges.
Where they briefly describe the most important data reported in the literature in a homogeneous manner and reinforce the relevance of Oxygen Nanocarriers as novel alternatives.
Thank you! The authors have made some modifications to clarify the aim of the study, and thus the novelty.
Authors should discuss whether the use of Oxygen Nanocarriers represents a solid alternative to existing therapeutics.
Thank you for the valuable suggestion! The authors have added some sentences, for example using the oxygen nanocarriers as cardioprotective agents.
Also, please discuss the use of method using green nanomaterials to targeting cells and mitochondria . Please add the below studies to your manuscript in the discussion section and bold your study novelties
-Ahmadov, I. S., et al. "The synthesis of the silver nanodrugs in the medicinal plant Baikal skullcap (Scutellaria Baicalensis Georgi) and their antioxidant, antibacterial activity." Advances in Biology & Earth Sciences 5.2 (2020): 103-118.
-Zhang, Junjie, et al. "Nanotechnological strategies to increase oxygen content of the tumor." Frontiers in Pharmacology 14 (2023): 646.
The authors highly acknowledge the reviewer for the valuable suggestions and thus have tried to modify the manuscript according to them. However, we hope that the aim of this study is clarified and the quality of the manuscript is enhanced.

Reviewer 3 Report
The manuscript reports the potential applications of two oligosaccharides (CNN and αCD) and their cross-linked polymers (CNN NS and αCD NS) as artifificial oxygen nanocarriers. The experimental results verified that oxygen-release kinetics were slower for CNN than for αCD , CNN NS and αCD NS displayed the ability to encapsulate, store and release oxygen into the cardioplegic solution for a prolonged period of time. These contents are interesting and fall within the scope of IJMS. I recommend its acceptance for publication after minor revision.
New strategies for delivering active ingredients are always highly desired, some recent new strategies can be cited and discussed in the INTRODUCTION section, such as https://doi.org/10.3390/nano13030551; https://doi.org/10.3390/molecules28062733; 10.1080/17425247.2023.2210834
In Table 2, please pay attention to the uniformity of significant digits.
The quanlity of Figure 2 to Figure 8 can be improved.
In the DISCUSSION section, the contents aboutthe synthesis mechanism of the CNN-based nanosponges can be more detailed.
Please pay attention to the uniformity of references’ formats and the ratio of the most recent three years’ references is too small.
can be improved!
Author Response
Manuscript ID: ijms-2417941
Dear Editor,
thank you for allowing us to submit a revised draft of the manuscript “Oxygen Nanocarriers for Improving Cardioplegic-Solution Performance: Physico-Chemical Characterization”. We appreciate the insightful feedback on our manuscript provided by the reviewers. We would like to thank the reviewers for the careful reading of this manuscript and for the thoughtful comments and constructive suggestions that will lead to valuable improvements in our paper. We have incorporated the suggestions made by the reviewers. They are highlighted within the manuscript by using “Track Changes” (red color) and below, please find the point-by-point responses to the comments and requests of the reviewers. The responses are presented in red.
Reviewer 3 Open Review
The manuscript reports the potential applications of two oligosaccharides (CNN and αCD) and their cross-linked polymers (CNN NS and αCD NS) as artifificial oxygen nanocarriers. The experimental results verified that oxygen-release kinetics were slower for CNN than for αCD , CNN NS and αCD NS displayed the ability to encapsulate, store and release oxygen into the cardioplegic solution for a prolonged period of time. These contents are interesting and fall within the scope of IJMS. I recommend its acceptance for publication after minor revision.
New strategies for delivering active ingredients are always highly desired, some recent new strategies can be cited and discussed in the INTRODUCTION section, such as https://doi.org/10.3390/nano13030551; https://doi.org/10.3390/molecules28062733; 10.1080/17425247.2023.2210834
Thank you very much for the positive feedback. We have tried to modify the entire manuscript, and therefore believe that the minor revision is reached.
In Table 2, please pay attention to the uniformity of significant digits.
Thank you for your valuable suggestion! The modifications are made.
The quanlity of Figure 2 to Figure 8 can be improved.
Thank you for the comment! The authors have tried to modify the Figures.
In the DISCUSSION section, the contents aboutthe synthesis mechanism of the CNN-based nanosponges can be more detailed.
Thank you for your valuable suggestion! The authors have added some more information regarding the mechanism of synthesized NSs.
Please pay attention to the uniformity of references’ formats and the ratio of the most recent three years’ references is too small.
Thank you very much for the observation! The required modifications are made.
Comments on the Quality of English Language
can be improved!
Thank you for the comment! The English language of the manuscript is entirely re-checked.

Reviewer 4 Report
This manuscript discuss about the Oxygen Nanocarriers for Improving Cardioplegic-Solution Performance: physico-chemical characterization. An interesting knowledge has been reported. However the following comments should be addressed before acceptance
minor comments
1. Novelty of the manuscript must be better emphasized
2. authors should remove the . in the introduction section
3. Suggested to add some qualitative results in the abstract section
4. keywords are not sound good, suggested to change the keywords
5. the importance and significance of this study should be more clearly disscussed with in the introduction section
6. Author have claimed " Nano-sponges are three-dimensional networks with remarkable adsorption properties due to their extensive nanometer-sized porosity. suggested to add more clear statement about their claim
7. authors should add more discussion about Figure 2. Oxygen diffusion from NIH-oxygen-saturated cardioplegic solutions at different temperatures.
8. How does sponge concentration influence the O2
9. How does the environmntal parametes effects nanoformulation preparations
10. there are some typological errors are present that should be revised carefully
After addressing all the comments this manuscript can be processed for further progress
There are some typological errors are present that should be carefully revised
Author Response
Manuscript ID: ijms-2417941
Dear Editor,
thank you for allowing us to submit a revised draft of the manuscript “Oxygen Nanocarriers for Improving Cardioplegic-Solution Performance: Physico-Chemical Characterization”. We appreciate the insightful feedback on our manuscript provided by the reviewers. We would like to thank the reviewers for the careful reading of this manuscript and for the thoughtful comments and constructive suggestions that will lead to valuable improvements in our paper. We have incorporated the suggestions made by the reviewers. They are highlighted within the manuscript by using “Track Changes” (red color) and below, please find the point-by-point responses to the comments and requests of the reviewers. The responses are presented in red.
Reviewer 4 Open Review
This manuscript discuss about the Oxygen Nanocarriers for Improving Cardioplegic-Solution Performance: physico-chemical characterization. An interesting knowledge has been reported. However the following comments should be addressed before acceptance
minor comments
- Novelty of the manuscript must be better emphasized
Thank you for the observation! We formulated two sentences in the Introduction and Discussion sections, and thus emphasizing the novelty of the work.
- authors should remove the . in the introduction section
Thank you! The introduction is entirely modified.
- Suggested to add some qualitative results in the abstract section
The authors thank the reviewer for the valuable suggestion. The entire abstract is modified and some qualitative results are mentioned.
- keywords are not sound good, suggested to change the keywords
Thank you for the comment! The authors have modified the keywords.
- the importance and significance of this study should be more clearly disscussed with in the introduction section
Thank you for the suggestion! There are added two paragraphs in the end of the Introduction section where it is described more the importance of the study.
- Author have claimed " Nano-sponges are three-dimensional networks with remarkable adsorption properties due to their extensive nanometer-sized porosity. suggested to add more clear statement about their claim
Thank you for your suggestion! The modifications in manuscript are made adding more information about the cyclodextrin-based nanosponges (CD-based NSs).
- authors should add more discussion about Figure 2. Oxygen diffusion from NIH-oxygen-saturated cardioplegic solutions at different temperatures.
Thank you for the observation! A clearer description is done!
- How does sponge concentration influence the O2
Thank you for the interesting question! It seems that the oxygen concentration increases as the concentration of nanoformulation increases at a certain quantities. As it can be observed in Table 5 the concentration of oxygen increases when the concentration of prepared nanosuspensions (αCD-NSs and CNN-NSs) increases from 0.13 g/L to 1.30 g/L. With further increase of the concentration at 13.0 g/L the concentration of oxygen decreases. This may be explained of the fact that there is a limited ratio between the nanosponge and oxygen. Furthermore, the influence of the nanosponge structure in the concentration of oxygen is confirmed when it is also compared with parent αCD and CNN (Table 5).
- How does the environmental parametes effects nanoformulation preparations
Thank you for your interesting question! A cardioplegic solution needs to be stable in storage and clinical use, simple to be prepared and used, preserve crucial heart functions and bears reasonable costs. Therefore, they are usually kept in the fridge to prevent any degradation.
- there are some typological errors are present that should be revised carefully
Thank you for the valuable observation! The authors have modified the entire manuscript.
After addressing all the comments this manuscript can be processed for further progress
Comments on the Quality of English Language
There are some typological errors are present that should be carefully revised
Thank you for the comment! The entire manuscript is grammarly re-checked, and thus we hope that the quality of the manuscript is enhanced.
